# Pneumonia-Masked Pulmonary Embolism in Nephrotic Syndrome: Diagnostic Value of V/Q Scintigraphy: A Case Report

**DOI:** 10.3390/reports8020042

**Published:** 2025-03-28

**Authors:** Ryosuke Saiki, Kan Katayama, Tomohiro Murata, Kaoru Dohi

**Affiliations:** Department of Cardiology and Nephrology, Mie University Graduate School of Medicine, Tsu 514-8507, Japan; katayamk@clin.medic.mie-u.ac.jp (K.K.); tmhr0421@med.mie-u.ac.jp (T.M.); dohik@med.mie-u.ac.jp (K.D.)

**Keywords:** nephrotic syndrome, pulmonary embolism, pulmonary infarction, ventilation–perfusion lung scintigraphy, case report

## Abstract

**Background and Clinical Significance:** Nephrotic syndrome predisposes patients to venous thromboembolism. This case highlights the challenges of diagnosing pulmonary embolism in nephrotic syndrome patients with renal dysfunction, and emphasizes the utility of ventilation–perfusion lung scintigraphy when the contrast is contraindicated. **Case Presentation:** A 52-year-old male presented with fatigue, left back pain, dyspnea, and lower limb edema. The laboratory findings indicated nephrotic syndrome with significant proteinuria, hypoalbuminemia, and impaired renal function. Elevated inflammatory markers and lung infiltrates on chest CT suggested pneumonia. Despite antibiotic therapy, lung shadows, and elevated D-dimer persisted. Lower extremity ultrasound was negative for deep vein thrombosis. Due to concerns about contrast-associated nephropathy, ventilation–perfusion lung scintigraphy was performed, revealing a right lung base mismatch, leading to a diagnosis of pulmonary embolism and infarction. A kidney biopsy confirmed minimal change in disease. The patient achieved complete remission of nephrotic syndrome and was discharged on oral anticoagulation. His oral anticoagulation was discontinued after 3 months due to sustained remission and the absence of deep vein thrombosis. **Conclusions:** Pulmonary embolism and infarction can occur even in the absence of deep vein thrombosis. ventilation–perfusion lung scintigraphy is useful for detecting pulmonary embolism in patients with impaired renal function.

## 1. Introduction and Clinical Significance

Venous thromboembolism (VTE) is a major complication of nephrotic syndrome (NS). NS is associated with an increased risk of VTE, with an estimated incidence of approximately 3.0% [1]. The incidence of pulmonary embolism, considered a severe complication, is even lower, occurring in 0.19% of patients with nephrotic syndrome [1]. In recent years, rapid developments in computed tomography (CT) technology have led to CT pulmonary angiography (CTPA) being the de facto clinical gold standard for the diagnosis of acute pulmonary embolism [2] due to its high sensitivity and specificity for pulmonary embolism, cost-effectiveness, and 24 h availability [2]. However, CTPA is often challenging to perform in patients with NS because NS is often complicated by acute kidney injury, and patients with acute kidney injury are often at high risk for contrast-associated nephropathy, so nephrologists are often hesitant to use contrast agents. Furthermore, IgG levels are known to decrease in nephrotic syndrome. Significantly, when the serum immunoglobulin G level falls below 600 mg/dL, and the serum creatinine level exceeds 2.0 mg/dL, the relative risk of bacterial infection increases to 5.31 [3]. This shows that the likelihood of other conditions that cause lung shadows, such as bacterial pneumonia, is elevated, as well as pulmonary embolism and pulmonary infarction, which makes diagnosis more difficult. For the risk assessment tool of pulmonary embolism, we can utilize the revised Geneva score [4] and the Wells clinical decision rule [5]. As readily tests, the D-dimer test and lower extremity ultrasound can be performed. D-dimer levels reflect the activation of coagulation and fibrinolysis, providing an assessment of thrombotic activity [6], and lower extremity ultrasound detects deep vein thrombosis (DVT), a precursor lesion of pulmonary embolism. Therefore, nephrologists sometimes rely on surrogate tests such as the D-dimer test and lower extremity venous ultrasound.

We encountered a patient with NS who presented with pulmonary shadows, but no deep vein thrombosis (DVT) was seen on lower extremity venous ultrasound, and a mismatch was seen on ventilation–perfusion (V/Q) lung scintigraphy, leading to a diagnosis of pulmonary embolism.

## 2. Case Presentation

A 52-year-old male was admitted to the hospital with NS and suspected pneumonia. He initially presented with fatigue and left back pain, followed by shortness of breath on exertion, and edema of the lower legs. In past health checkups, only mild dyslipidemia and hyperuricemia were noted, and the patient was not on any regular medication. His height was 176.6 cm, and his weight was 89 kg, representing a 6 kg increase from his usual body weight. A physical examination at the time of admission revealed the following: blood pressure, 136/81 mmHg; heart rate, 86 beats/min; body temperature, 36.7 °C; and oxygen saturation, 97% on room air. Coarse crackles were auscultated in his bilateral lower lung fields, and severe left chest pain was noted upon inspiration. Pitting edema was observed in the lower extremities, but there was no sign of heat or redness.

The laboratory data are shown in (Table 1). Blood and urine tests revealed substantial proteinuria (11.26 g/day) and markedly low serum albumin (0.7 g/dL), meeting the criteria for nephrotic syndrome. Renal function was impaired, as demonstrated by an elevated serum creatinine (1.62 mg/dL) and a reduced estimated glomerular filtration rate (36.8 mL/min/1.73 m^2^). Inflammatory markers were also elevated, with a C-reactive protein level of 22.11 mg/dL and procalcitonin of 0.5 ng/mL. D-dimer was elevated at 7.7 μg/mL.

An electrocardiogram demonstrated a normal axis with no significant abnormalities in the QRST complexes. Echocardiography did not reveal any abnormalities. Based on the presence of lung infiltration (Figure 1A,B), bacterial pneumonia and pleuritis were suspected, and antibiotic treatment was initiated. In addition, given the presence of acute kidney injury and pulmonary infiltrates, the possibility of anti-glomerular basement membrane (GBM) disease and antineutrophil cytoplasmic antibody (ANCA)-associated vasculitis needed to be considered. Although prophylactic anticoagulation therapy would typically be initiated from the start of treatment in cases of severe nephrotic syndrome, caution was necessary due to the risk of alveolar hemorrhage caused by anti-GBM disease or ANCA-associated vasculitis. Consequently, prophylactic anticoagulation therapy was only initiated after confirming the absence of markers relevant to these conditions. Lower extremity venous ultrasound did not reveal any evidence of DVT. Although the patient exhibited no fever and showed improvement in C-reactive protein levels based on blood test results, indicating that the infection appeared to be under control, follow-up CT scans on the ninth day of hospitalization revealed only a slight improvement in pulmonary shadows. D-dimer was still elevated at 10.37 μg/mL.

This finding necessitated consideration of causes other than bacterial infections. Although DVT was not observed, it was deemed necessary to rule out pulmonary embolism. Considering the risk of contrast-associated nephropathy, contrast-enhanced CT was avoided, and ventilation–perfusion (V/Q) lung scintigraphy was conducted on the sixteenth day of hospitalization, revealing a mismatch in the base of the right lung (Figure 2A,B), leading to the diagnosis of right pulmonary embolism and infarction. A follow-up CT scan was conducted, revealing that the pulmonary shadow had decreased in size, making the previously concealed wedge-shaped shadow more clearly visible (Figure 1C). It is believed that the improvement in bacterial pneumonia allowed the pulmonary infarction to become clearly visible. Positron emission tomography-computed tomography was performed to evaluate for a tumor, and coagulation factor markers, including lupus anticoagulant, anti-cardiolipin-beta2-glycoprotein I complex antibody, anti-cardiolipin immunoglobulin G, and anti-cardiolipin immunoglobulin M, were evaluated to check for potential coagulation system disorders. However, no specific abnormalities were identified. The etiology of the patient’s pulmonary embolism and infarction was considered to be attributed to a hypercoagulable state associated with NS.

Regarding NS, a kidney biopsy was performed immediately after hospitalization before the start of anticoagulant therapy, and a histological examination of 11 glomeruli revealed no evidence of global sclerosis, segmental sclerosis, adhesions, or crescent formation (Figure 3A). Swelling and vacuolization were observed in the proximal tubular epithelial cells (Figure 3B). Immunofluorescence did not reveal any specific changes, while electron microscopy showed extensive foot process effacement without any electron-dense deposits (Figure 3C). Based on these findings, the patient was diagnosed with minimal change disease and acute tubular necrosis. Although he was treated with prednisolone (1 mg/kg), he had oliguria and his creatinine levels deteriorated to 7.33 mg/dL, which needed hemodialysis treatment. Despite the need for temporary blood purification therapy, the patient successfully discontinued dialysis, and complete remission of the NS was achieved. The patient was subsequently discharged from the hospital. Oral anticoagulation was achieved with warfarin. He did not complain of any symptoms, such as shortness of breath, chest pain, or leg edema after being discharged from the hospital. As mentioned previously, the pulmonary embolism was considered to be provoked by nephrotic syndrome. Anticoagulant therapy was discontinued after 3 months because the patient continued to show complete remission of nephrotic syndrome.

## 3. Discussion

We reiterated two clinical issues. Pulmonary embolism and infarction frequently occur even in the absence of DVT. V/Q lung scintigraphy is useful for detecting pulmonary embolism in patients with impaired renal function.

First, pulmonary embolism and infarction frequently occur even in the absence of DVT. When feasible, we aim to diagnose pulmonary embolism and infarction using CTPA, which is considered the gold standard for diagnosis [2]. However, there are several dilemmas in the process of diagnosing pulmonary embolism and infarction in patients with renal impairment. A significant drawback of CTPA is its potential to negatively impact renal function. Contrast-associated nephropathy occurs in less than 1% of the general population but increases to approximately 15% in high-risk groups, including patients with kidney injury [7]. Normal saline is commonly used for the prevention of contrast-associated nephropathy and is frequently used as a control group in comparative trials of many new drugs [8,9]. In addition to normal saline, past systematic reviews have reported that low-osmolar contrast media, N-acetylcysteine, and statin medications are effective as preventive treatments for contrast-associated nephropathy [8,9]. This indicates that saline is a critical agent for preventing contrast-associated nephropathy. However, administering saline to patients with acute kidney injury who exhibit hypervolemia and pleural effusion poses a significant risk of severe complications, including respiratory failure. As simple tests related to DVT and pulmonary embolism, there are D-dimer tests and lower extremity venous ultrasound examinations. In patients with a low pretest probability and a negative D-dimer test result, pulmonary embolism and infarction can be excluded without the need for additional imaging studies, but D-dimer levels generally increase in response to deterioration of the renal function [10] because D-dimer is cleared by the kidneys and the reticuloendothelial system [6]. D-dimer testing cannot be effectively utilized in patients with certain conditions, such as renal impairment. In addition, studies have reported that approximately 46% of patients with pulmonary embolism have negative DVT findings on ultrasonography [11]. Consequently, lower-extremity ultrasonography results are also not sufficient to rule out pulmonary embolism or infarction. In retrospect, the diagnosis in this case was established 16 days post-admission. However, given the potentially life-threatening nature of pulmonary embolism, if any suspicion existed, the imaging studies should have been expedited. The guidelines do not indicate that CTPA is an absolute contraindication in acute kidney injury patients [12]. In this case, it was considered desirable to perform contrast-enhanced CT, accepting the risk of renal impairment, or to perform an alternative examination such as ventilation–perfusion lung scintigraphy.

Second, V/Q lung scintigraphy is useful for detecting pulmonary embolism in patients with impaired renal function. Although some had argued in the past that the sensitivity and specificity of V/Q scintigraphy were problematic [13], it is an option for nephrologists concerned with impaired renal function as it does not lead to contrast-associated nephropathy [14]. Lung ventilation/perfusion scintigraphy is a useful tool for evaluating patients with nephrotic syndrome to detect pulmonary embolism [15]. The major weakness of V/Q scintigraphy lies in its alarmingly high rate of non-diagnostic results, reaching around 50% [16], a figure that is further exacerbated, leading to decreased accuracy, in the presence of pulmonary infiltrates or opacities on chest imaging [17]. When using V/Q scintigraphy, it is necessary to pay attention to this point.

The present report described the case of a patient with NS complicated by pneumonia, pulmonary embolism, and infarction. The diagnosis took slightly longer because the lung infarction shadow was obscured by pneumonia findings. Patients with NS frequently develop not only concurrent pulmonary embolism or infarction but also infections. This case has taught us the importance of suspecting a pulmonary infarction hidden behind pneumonia. Pulmonary embolism is a potentially fatal condition. Even patients who appear stable succumb to death from pulmonary embolism [18], emphasizing the importance of constant vigilance in managing this condition.

## 4. Conclusions

In conclusion, pulmonary embolism and infarction frequently occur even in the absence of DVT, and V/Q lung scintigraphy is an option for detecting pulmonary embolism in patients with impaired renal function.

When a shadow is observed in the lungs of a patient with NS, even if pneumonia is suspected, it is crucial to consider the possibility of a pulmonary embolism or infarction. Importantly, these conditions cannot be ruled out solely based on lower limb ultrasound or D-dimer tests. If a decrease in oxygen saturation occurs that cannot be explained by pneumonia alone, or if test abnormalities such as a gradual increase in D-dimer levels are present, further detailed investigation is necessary.

While contrast-enhanced CT is the gold standard for diagnosis, V/Q scintigraphy remains a useful diagnostic option, in situations where the use of contrast agents is hesitated.

## Figures and Tables

**Figure 1 reports-08-00042-f001:**
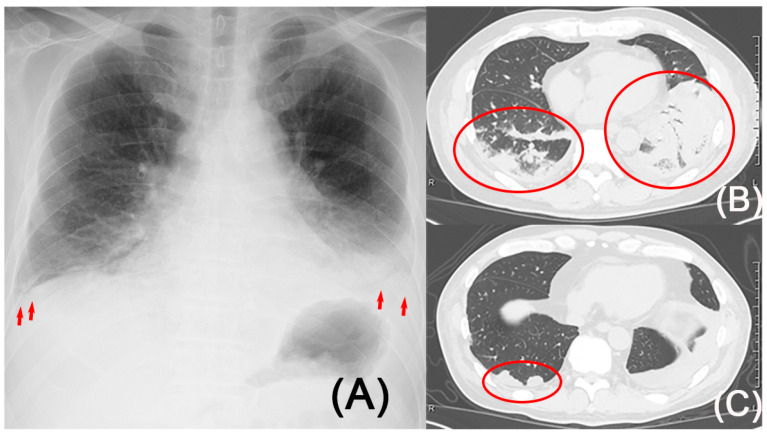
Radiography and computed tomography. (**A**) Radiography revealed bilateral lower lung field opacities, pleural effusion (arrow), and cardiomegaly. (**B**) Computed tomography demonstrated bilateral lower lobe consolidations (circle). (**C**) Computed tomography showed a reduction in bilateral lung opacities and revealed a wedge-shaped opacity (circle) in the right lower lung field.

**Figure 2 reports-08-00042-f002:**
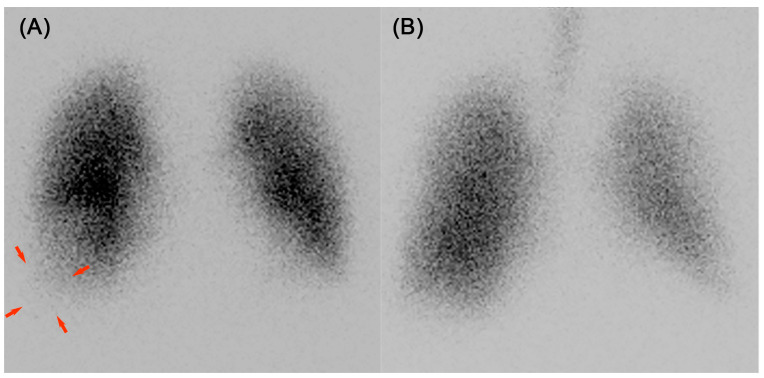
Ventilation–perfusion lung scintigraphy: (**A**) perfusion and (**B**) ventilation. A ventilation–perfusion lung scintigraphy demonstrated a mismatch in the right lung base, which was captured in the anterior view. Arrows indicated the missing areas.

**Figure 3 reports-08-00042-f003:**
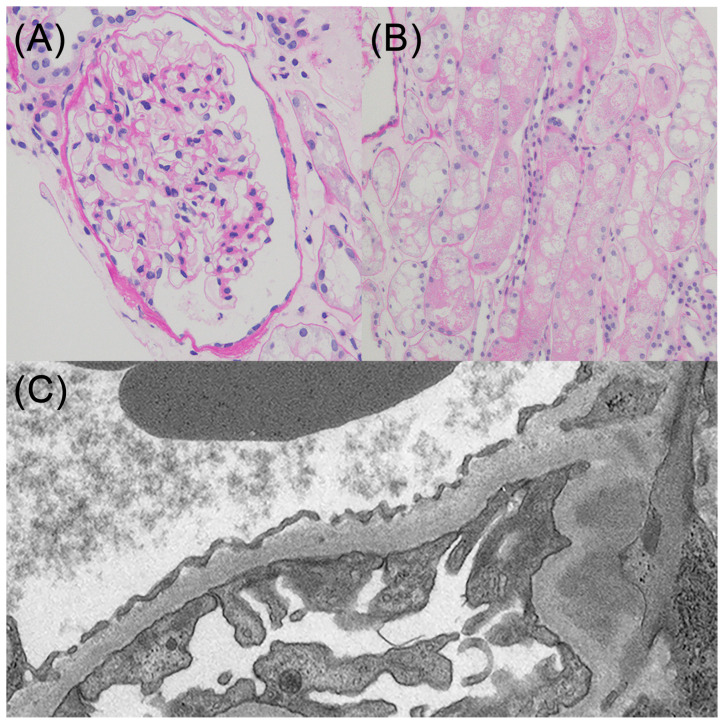
Kidney biopsy (**A**) The representative glomerulus. There was no evidence of global sclerosis, segmental sclerosis, adhesions, or crescent formation. (**B**) Swelling and vacuolization were observed in the proximal tubular epithelial cells. (**C**) Electron microscopy showed extensive foot process effacement without any electron-dense deposits.

**Table 1 reports-08-00042-t001:** Laboratory data.

Name	Values	Unit	Normal Range
pH	6	NA	4.5–7.5
Protein	11.26	g/day	NA
RBC	1–4	/HPF	<5
NAG	120.8	IU/L	1.0–4.2
BJP	(-)	NA	(-)
WBC	11,130	/μL	3300–8600
RBC	588	×10^4^/μL	435–555
Hb	18.3	g/dL	13.7–16.8
Plt	21.2	×10^4^/μL	15.8–34.8
ANA	1:40	NA	NA
MPO-ANCA	<0.2	IU/mL	0–3.5
PR3-ANCA	<0.6	IU/mL	0–2.0
Anti-GBM	<1.5	IU/mL	0–7.0
ds-DNA	<0.6	IU/mL	0–10.0
Procalcitonin	0.5	ng/mL	0–0.05
D-dimer	7.7	μg/mL	0–1.0
HbA1c	5.5	%	4.9–6.0
Glu	78	mg/dL	73–109
TP	4.3	g/dL	6.6–8.1
Alb	0.7	g/dL	4.1–5.1
BUN	31.9	mg/dL	8.0–20.0
Cr	1.62	mg/dL	0.65–1.07
eGFR	36.8	mL/min/1.73 m^2^	NA
Na	129	mEq/L	138–145
K	4.7	mEq/L	3.6–4.8
Cl	98	mEq/L	101–108
AST	30	U/L	13–30
ALT	14	U/L	10–42
LDH	415	U/L	124–222
ALP	114	U/L	38–113
γGTP	101	U/L	13–64
CRP	22.11	mg/dL	0–0.14
IgG	485	mg/dL	861–1747
IgA	407	mg/dL	93–393
IgM	100	mg/dL	33–183
C3	195	mg/dL	73–138
C4	57.1	mg/dL	11–31
CH50	56.2	U/mL	31.6–57.6

Alb, albumin; ALP, alkaline phosphatase; ALT, alanine transaminase; ANA, antinuclear antibody; Anti-GBM, anti-glomerular basement membrane antibody; AST, asparate transaminase; BJP, bence jones protein; BUN, blood urea nitrogen; C3, complement 3; C4, complement 4; CH50, 50% hemolytic complement activity; Cl, chloride; Cr, creatinine; CRP, c-reactive protein; ds-DNA, double-stranded deoxyribonucleic acid antibody; eGFR, estimated glomerular filtration rate; γGTP, γ-glutamyltranspeptidase; Hb, hemoglobin; HbA1c, hemoglobin a1c; HBV, hepatitis B virus; HCV, hepatitis C virus; HIV, human immunodeficiency virus; IgA; immunoglobulin A, IgG, immunoglobulin G; IgM; immunoglobulin M; K, kalium; LDH, lactate dehydrogenase; MPO-ANCA, myeloperoxidase antineutrophil cytoplasmic antibody; Na, natrium; NA, not applicable; NAG, N-acetyl-β-D-glucosaminidase; Plt, platelets; PR3-ANCA, proteinase3-antineutrophil cytoplasmic antibody; RBC, red blood cells; TP, total protein; WBC, white blood cells.

## Data Availability

The original data presented in this study are available on reasonable request from the corresponding author. The data are not publicly available due to privacy concerns.

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
