# Peer review of "Pneumonia-Masked Pulmonary Embolism in Nephrotic Syndrome: Diagnostic Value of V/Q Scintigraphy: A Case Report"

_reports, 2025, doi:10.3390/reports8020042_

Round 1

Reviewer 1 Report

Comments and Suggestions for Authors

This is an interesting case of patient with nephrotic syndrome, acute kidney injury and subsequent pulmonary embolism that was diagnosed with lung scintigraphy as CTPA was chosen not to be used due to the concurrent AKI.

Authors overemphasize the relative contraindication of the use of i.v. iodinated contrast for the diagnosis of pulmonary embolism in patients with AKI. Nevertheless, it should be noted that the presence of AKI or lowered eGFR (commonly < 30 ml/min/1.73m2) is only a relative contraindication and not an absolute one as stated in KDIGO and other guidelines.

Authors state in discussion that CIN occurs in about 15% of high risk patients. Although this is based on data from the study of Mamoulakis et al, there are concerns about the correct terminology and the data used for this study. For instance the risk for CIN is different between intra-venous and intra-arterial use of iodinated contrast media that were mentioned together in that study. As there is much concern for even the existence of ‘’pure’’ CIN, in my opinion the use of the term ‘’contrast associated nephropathy’’ would be more proper in this case report. Furthermore, as suggested in JAMA Intern Med. 2021;181(6):767-74 the use of contrast media for CTPA in the emergency department is not associated with AKI. Overall, the use of scintigraphy over CTPA should not be over-emphasized in this case report as the actual existence of CI-AKI is not fully validated.

Author Response

Comments 1: 
Authors overemphasize the relative contraindication of the use of i.v. iodinated contrast for the diagnosis of pulmonary embolism in patients with AKI. Nevertheless, it should be noted that the presence of AKI or lowered eGFR (commonly < 30 ml/min/1.73m2) is only a relative contraindication and not an absolute one as stated in KDIGO and other guidelines.

Response 1: Thank you for pointing this out. We agree with this comment. Therefore, we have replaced "the aforementioned imaging" with "the imaging" on line 190. I have also removed lines 190-197, which included the sentences starting with "Furthermore, if clinically warranted..." and ending with "...risks compromising renal function." and added in that section: "The guidelines do not indicate that CTPA is an absolute contraindication in acute kidney injury patients [12]. In this case, it was considered desirable to perform contrast-enhanced CT, accepting the risk of renal impairment, or to perform an alternative examination such as ventilation-perfusion lung scintigraphy."

Comments 2: Authors state in discussion that CIN occurs in about 15% of high risk patients. Although this is based on data from the study of Mamoulakis et al, there are concerns about the correct terminology and the data used for this study. For instance the risk for CIN is different between intra-venous and intra-arterial use of iodinated contrast media that were mentioned together in that study. As there is much concern for even the existence of ‘’pure’’ CIN, in my opinion the use of the term ‘’contrast associated nephropathy’’ would be more proper in this case report. Furthermore, as suggested in JAMA Intern Med. 2021;181(6):767-74 the use of contrast media for CTPA in the emergency department is not associated with AKI. Overall, the use of scintigraphy over CTPA should not be over-emphasized in this case report as the actual existence of CI-AKI is not fully validated.

Response 2: We thank the reviewer for their insightful comments and careful reading of our case report. We have revised the manuscript to consistently use the term "contrast-associated nephropathy," as suggested, to reflect the potential association without implying definite causality. Finally, lines 192-194. We revised it: In this case, it was considered desirable to perform contrast-enhanced CT, accepting the risk of renal impairment, or to perform an alternative examination such as ventilation-perfusion lung scintigraphy. We wrote it in parallel.

Reviewer 2 Report

Comments and Suggestions for Authors

Remarks

The article needs a major revision. More detailed remarks are mentioned below.

Line 50 -

For risk stratification of pulmonary embolism, we can utilize the revised Geneva score [4] and the Wells clinical decision rule [5]. As readily available alternative diagnostic tests, the D-dimer test and lower extremity ultrasound can be performed.

This is not risk stratification  but risk assessment tool.

Alternative diagnostic test – this statement is wrong one. Please, look at the flowchart of diagnosis of suspected pulmonary embolism. Please, look at the discussion part – it is different.

References 4 and 5 as  - do not discuss risk stratification but risk assessment.

D-dimer levels reflect the activation of coagulation and fibrinolysis, providing a swift assessment of thrombotic activity [6)

I do not understand swift assessment. D-dimer »production« is »normal« in humans, but it could be increased in a lot of diseases – so, only a sub-threshold value is important to exclude PE (in low-risk patients)

Line 58

We encountered a patient with NS who presented with pulmonary shadows; D-dimer was elevated,

D-dimer is useless as a PE diagnostic tool in lung infection,…

Line 119

, revealing a mismatch in the base of the right lung (Figure 119 2A, 2B),

I wonder what the result on the left side was? Probably some changes, too?

Line 135

Regarding NS, a kidney biopsy was performed immediately after hospitalization, and histological examination of 11 glomeruli revealed no evidence of global sclerosis, segmental sclerosis, adhesions, or crescent formation (Figure 3A

Few words about anticoagulation and biopsy – bleeding risk?, biopsy done before anticoagulation?

Line 146

Oral anticoagulation was achieved with warfarin. He did not complain of any symptoms such as shortness of breath, chest pain, or leg edema after being discharged from the hospital. Anticoagulant therapy was discontinued after 3 months because the patient remained free of lower extremity DVT and continued to show complete remission of nephrotic syndrom.

Longer treatment of pulmonary embolism (PE) (or venous thromboembolic event) is not longer or prolonged due to the symptoms or signs but due to prevent the recurrency. So, maybe some words about provoked or unprovoked PE?

Line 157

Pulmonary embolism and infarction can occur even in the absence of DVT

This happens in about 40% of PE –it is not a rare event –  also mentioned in the discussion section

Line 165

Contrast-induced nephropathy (CIN) occurs in less than 1% of the general population but increases to approximately 15% in high-risk groups, including patients with acute kidney injury [7].

In the cited reference, I did not find the contrast use in acute renal failure. I agree that this is plausible in NS patients.

However, CIN data with the new agents in lower quantities and prophylactic treatment (just fluids) are more encouraging than 15 %.

line 168

The efficacy of these preventive measures varies, with previous systematic reviews and metaanalyses demonstrating their effectiveness when low-osmolar contrast media is used in conjunction with intravenous administration of low-dose N-acetylcysteine and saline, or a combination therapy involving statins, N-acetylcysteine, and saline [8]

There new data about these prophylacitc measures. Maybe a new reference?

Line 176

Therefore, nephrologists use alternative tests, D-dimer test and lower extremity venous ultrasound, to assess pulmonary embolism and infarction when the pre-test probability is not so high

D-dimer is used as risk assessment tool – so, before the other objective tests are used. It is used in low clinical probability only as it is correctly pointed out in the next sentence.

Line 182

D-dimer testing cannot be effectively utilized in patients with renal impairment.

True, but it can not be used in a lot of diseases where it is increased.

Line 186

While research indicates significant differences in fibrinogen and antithrombin III levels between nephrotic syndrome patients with and without pulmonary embolism [11].

What follows – why this statement

Line 200

, it is an appealing test for nephrologists concerned with impaired renal function as it does not lead to contrast-induced nephropathy [13]. Lung ventilation/perfusion scintigraphy proved a useful tool for evaluating patients with nephrotic syndrome, providing more information than lower extremity edema, chest radiography, or electrocardiogram alone [11].

The problem of V/Q scintigraphy is the accuracy, because it is not precise enough (only in about 50% - and when there are some changes in the lungs (as in this patient) the accuracy goes down.

Line 202

Lung ventilation/perfusion scintigraphy proved a useful tool for evaluating patients with nephrotic syndrome, providing more information than lower extremity edema, chest radiography, or electrocardiogram alone [11]

V/Q scintigraphy is certainly better than the methods mentioned above, which are methods not really used in diagnosing PE.

Line 212

so MRPA and V/Q SPECT were not per- 212

formed. One challenge is that access to V/Q scintigraphy, non-contrast MRPA, and V/Q

SPECT can be limited in certain settings, particularly in non-academic facilities [17]. Additionally, molybdenum-99 and technetium-99m are not suitable for storage due to their brief half-lives, and prolonged, unanticipated reactor shutdowns have caused serious shortages of these isotopes in several countries [17]. In recent years, a technology called dynamic chest radiography, which applies conventional X-ray equipment, has also been developed for detecting pulmonary embolism [17]. This specialized system requires an optimal combination of a flat-panel detector and a pulsed X-ray generator, along with a high-performance image processing workstation [17]. However, the initial installation cost is estimated to be one-tenth that of a SPECT/CT system [17], making it a relatively affordable option. This report focuses on chronic pulmonary embolism [17], but with future advancements in technology, the challenges associated with the use of contrast agents and issues related to nuclear medicine may eventually be addressed

I do not understand why a discussion about the method not used, but no discussion about the used –one (planar V/Q sc)

Line 226

The present report described the case of a patient with NS complicated by pneumonia, pulmonary embolism, and infarction

Please, elaborate on pneumonia – does he had it?

Line 244

As additional tests, in cases presenting with acute kidney injury, tests such as V/Q

scintigraphy, V/Q SPECT, and MRPA should be considered. However, these tests can only be performed at a limited number of hospitals, and even at those hospitals, the time required to conduct these tests varies. If necessary for various reasons, contrast-enhanced CT should also be performed

After reading this paragraph, it is not possible to understand which method for diagnosing PE should be used firstly. I agree that there is not test that is 100% accurate, but CTPA is the gold standard as it is also mentioned in this article.

Author Response

Comments 1:
Line 50 -
For risk stratification of pulmonary embolism, we can utilize the revised Geneva score [4] and the Wells clinical decision rule [5]. As readily available alternative diagnostic tests, the D-dimer test and lower extremity ultrasound can be performed. 
This is not risk stratification but risk assessment tool.
Alternative diagnostic test – this statement is wrong one. Please, look at the flowchart of diagnosis of suspected pulmonary embolism. Please, look at the discussion part – it is different.
References 4 and 5 as  - do not discuss risk stratification but risk assessment.
D-dimer levels reflect the activation of coagulation and fibrinolysis, providing a swift assessment of thrombotic activity [6)
I do not understand swift assessment. D-dimer »production« is »normal« in humans, but it could be increased in a lot of diseases – so, only a sub-threshold value is important to exclude PE (in low-risk patients)
Response 1: 
As suggested, we changed sentences in line 50-55 into "For risk assessment tool of pulmonary embolism, we can utilize the revised Geneva score [4] and the Wells clinical decision rule [5]. As readily tests, the D-dimer test and lower extremity ultrasound can be performed. D-dimer levels reflect the activation of coagula-tion and fibrinolysis, providing an assessment of thrombotic activity [6], and lower ex-tremity ultrasound detects deep vein thrombosis (DVT), a precursor lesion of pulmonary embolism." .

Comments 2: 
Line 58
We encountered a patient with NS who presented with pulmonary shadows; D-dimer was elevated,
D-dimer is useless as a PE diagnostic tool in lung infection,…
Response 2: 
Thank you for your comment regarding line 58. You are correct that elevated D-dimer is often non-specific in the setting of pulmonary infection and has limited diagnostic utility for pulmonary embolism in this context. To improve the clarity of the sentence, We have removed the phrase mentioning the elevated D-dimer.

Comments 3:
Line 119
, revealing a mismatch in the base of the right lung (Figure 119 2A, 2B),
I wonder what the result on the left side was? Probably some changes, too?
Line 135
Regarding NS, a kidney biopsy was performed immediately after hospitalization, and histological examination of 11 glomeruli revealed no evidence of global sclerosis, segmental sclerosis, adhesions, or crescent formation (Figure 3A
Few words about anticoagulation and biopsy – bleeding risk?, biopsy done before anticoagulation?
Response 3: 
Thank you for your feedback. No significant findings, such as mismatch, were observed in the left lung.
As you pointed out, the chronological relationship with anticoagulant therapy was unclear, so I have made the following revisions.
Regarding NS, a kidney biopsy was performed immediately after hospitalization before the start of anticoagulant therapy, and histological examination of 11 glomeruli revealed no evidence of global sclerosis, segmental sclerosis, adhesions, or crescent formation (Figure 3A).

Comments 4: 
Line 146

Oral anticoagulation was achieved with warfarin. He did not complain of any symptoms such as shortness of breath, chest pain, or leg edema after being discharged from the hospital. Anticoagulant therapy was discontinued after 3 months because the patient remained free of lower extremity DVT and continued to show complete remission of nephrotic syndrom.

Longer treatment of pulmonary embolism (PE) (or venous thromboembolic event) is not longer or prolonged due to the symptoms or signs but due to prevent the recurrency. So, maybe some words about provoked or unprovoked PE?
Response 4: 
Thank you for your comment. We changed "Anticoagulant therapy was discontinued after 3 months because the patient remained free of lower extremity DVT and continued to show complete remission of nephrotic syn-drome." into "As mentioned previously, the pulmonary embolism was considered to be provoked by nephrotic syndrome. Anticoagulant therapy was discontinued after 3 months because the patient continued to show complete remission of nephrotic syndrom." in line 147-150.

Comments 5: 
Line 157

Pulmonary embolism and infarction can occur even in the absence of DVT

This happens in about 40% of PE –it is not a rare event –  also mentioned in the discussion section
Response 5: 
As you pointed out, we changed 'can' to 'frequently' on lines 157 and 160. We also changed 'can' to 'frequently' in the first sentence of the conclusion.

Comments 6: 
Line 165

Contrast-induced nephropathy (CIN) occurs in less than 1% of the general population but increases to approximately 15% in high-risk groups, including patients with acute kidney injury [7].

In the cited reference, I did not find the contrast use in acute renal failure. I agree that this is plausible in NS patients.

However, CIN data with the new agents in lower quantities and prophylactic treatment (just fluids) are more encouraging than 15 %.

Response 6: 
As you pointed out, there was no mention of 'acute kidney injury'. I apologize for the oversight. I have removed 'acute' from the text and changed it to 'kidney injury'.

As suggested, we changed “Preventive medications for contrast-associated nephropathy include saline, N-acetylcysteine, sodium bicarbonate, ascorbic acid, and statins [8]. The efficacy of these preventive measures varies, with previous systematic reviews and meta-analyses demonstrating their effectiveness when low-osmolar contrast media is used in conjunction with intravenous administration of low-dose N-acetylcysteine and saline, or a combination therapy involving statins, N-acetylcysteine, and saline [8].” into “Normal saline is commonly used for the prevention of contrast-associated nephropathy and is frequently used as a control group in comparative trials of many new drugs [8,9]. In addition to normal saline, past systematic reviews have reported that low-osmolar contrast media, N-acetylcysteine, and statin medications are effective as preventive treatments for contrast-associated nephropathy [8,9].” in line 167-171.

Comments 7: 
line 168

The efficacy of these preventive measures varies, with previous systematic reviews and metaanalyses demonstrating their effectiveness when low-osmolar contrast media is used in conjunction with intravenous administration of low-dose N-acetylcysteine and saline, or a combination therapy involving statins, N-acetylcysteine, and saline [8]

There new data about these prophylacitc measures. Maybe a new reference?

Response 7: 
In response to your suggestion, we have added a new reference, reference number 9.

Comments 8: 
Line 176

Therefore, nephrologists use alternative tests, D-dimer test and lower extremity venous ultrasound, to assess pulmonary embolism and infarction when the pre-test probability is not so high

D-dimer is used as risk assessment tool – so, before the other objective tests are used. It is used in low clinical probability only as it is correctly pointed out in the next sentence.

Response 8: 
Thank you for the insightful comment. We changed "Therefore, nephrologists use alternative tests, D-dimer test and lower extremity venous ultrasound, to assess pulmonary embolism and infarction when the pre-test probability is not so high. " into "As simple tests related to DVT and pulmonary embolism, there are D-dimer tests and lower extremity venous ultrasound examinations.".  in line 176-177.

Comments 9: 
Line 182
D-dimer testing cannot be effectively utilized in patients with renal impairment.
True, but it can not be used in a lot of diseases where it is increased.
Response 9: 
Thank you for the insightful comment. We added “certain conditions, such as” into the sentence in line 182.

Comments 10: 
Line 186

While research indicates significant differences in fibrinogen and antithrombin III levels between nephrotic syndrome patients with and without pulmonary embolism [11].

What follows – why this statement

Response 10: 
Regarding line 186, We have removed the sentence. The original intent of including the sentence which is “While research indicates significant differences in fibrinogen and antithrombin III levels between nephrotic syndrome patients with and without pulmonary embolism [11].” was not clear, and it disrupted the flow and focus of the surrounding text. It did not lead logically into the subsequent discussion.

Comments 11: 
Line 200

, it is an appealing test for nephrologists concerned with impaired renal function as it does not lead to contrast-induced nephropathy [13]. Lung ventilation/perfusion scintigraphy proved a useful tool for evaluating patients with nephrotic syndrome, providing more information than lower extremity edema, chest radiography, or electrocardiogram alone [11].

The problem of V/Q scintigraphy is the accuracy, because it is not precise enough (only in about 50% - and when there are some changes in the lungs (as in this patient) the accuracy goes down.

Response 11: 
Thank you for pointing out the inaccuracy regarding V/Q scintigraphy. We have revised the sentence to:

"...it is an option for nephrologists concerned with impaired renal function as it does not lead to contrast-associated nephropathy."

We acknowledge the limitations of V/Q scintigraphy's accuracy, and appreciate your highlighting this important point.

Comments 12: 
Line 202

Lung ventilation/perfusion scintigraphy proved a useful tool for evaluating patients with nephrotic syndrome, providing more information than lower extremity edema, chest radiography, or electrocardiogram alone [11]

V/Q scintigraphy is certainly better than the methods mentioned above, which are methods not really used in diagnosing PE.
Response 12: 
Thank you for your feedback. We reduced and adjusted the degree of recommendation about V/Q scintigraphy. We believe these changes address your concerns and provide a more accurate representation of the current clinical understanding of V/Q scintigraphy.

Comments 13: 
Line 212

so MRPA and V/Q SPECT were not per- 212

formed. One challenge is that access to V/Q scintigraphy, non-contrast MRPA, and V/Q

SPECT can be limited in certain settings, particularly in non-academic facilities [17]. Additionally, molybdenum-99 and technetium-99m are not suitable for storage due to their brief half-lives, and prolonged, unanticipated reactor shutdowns have caused serious shortages of these isotopes in several countries [17]. In recent years, a technology called dynamic chest radiography, which applies conventional X-ray equipment, has also been developed for detecting pulmonary embolism [17]. This specialized system requires an optimal combination of a flat-panel detector and a pulsed X-ray generator, along with a high-performance image processing workstation [17]. However, the initial installation cost is estimated to be one-tenth that of a SPECT/CT system [17], making it a relatively affordable option. This report focuses on chronic pulmonary embolism [17], but with future advancements in technology, the challenges associated with the use of contrast agents and issues related to nuclear medicine may eventually be addressed

I do not understand why a discussion about the method not used, but no discussion about the used –one (planar V/Q sc)

Response 13: 
Thank you for pointing out the lack of discussion regarding planar V/Q scintigraphy. We agree that the discussion should focus more on the method used. We will revise the paragraph to address this. Specifically, we will remove the sentence "In this case, pulmonary embolism was diagnosed by the initial V/Q scintigraphy, so MRPA and V/Q SPECT were not performed" and add the following: "The major weakness of V/Q scintigraphy lies in its alarmingly high rate of non-diagnostic results, reaching around 50%. When using V/Q scintigraphy, it is necessary to pay attention to this point."

Comments 14: 
Line 226

The present report described the case of a patient with NS complicated by pneumonia, pulmonary embolism, and infarction

Please, elaborate on pneumonia – does he had it?
Response 14: 
Thank you for your inquiry. As clearly stated in the case report, we consider this to be a case of bacterial pneumonia complicated by pulmonary embolism and infarction. We showed that the improvement in bacterial pneumonia allowed the pulmonary infarction to become clearly visible in case part in line 119-121.

Comments 15: 
Line 244

As additional tests, in cases presenting with acute kidney injury, tests such as V/Q

scintigraphy, V/Q SPECT, and MRPA should be considered. However, these tests can only be performed at a limited number of hospitals, and even at those hospitals, the time required to conduct these tests varies. If necessary for various reasons, contrast-enhanced CT should also be performed

After reading this paragraph, it is not possible to understand which method for diagnosing PE should be used firstly. I agree that there is not test that is 100% accurate, but CTPA is the gold standard as it is also mentioned in this article.

Response 15: 
Thank you for your comment. We have revised the paragraph on line 244 as suggested. We have removed the original text and replaced it with: "While contrast-enhanced CT is the gold standard for diagnosis, V/Q scintigraphy remains a valid diagnostic option, particularly in cases where renal dysfunction or other factors preclude the use of contrast agents." Additionally, we have changed "useful" to "an option" on line 231 as you suggested.

Round 2

Reviewer 1 Report

Comments and Suggestions for Authors

Authors have adequately responded to my points of consideration.

Author Response

Thank you for your careful review of our manuscript. 

Reviewer 2 Report

Comments and Suggestions for Authors

Author Response

Comment1:
Line 71 
The below abreviations are legend and not the text – please shape it, I would omit so manj (unimportant tests)- they make the artcle less Alb, albumin; ALP, alkaline phosphatase; ALT, alanine transaminase; ANA, antinu- 79 clear antibody; Anti-GBM, anti-glomerular basement membrane antibody; ASO, anti- 80 streptolysin O antibody; AST, asparate transaminase; β2MG, β2-microglobulin; BJP, bence 81 jones protein; BUN, blood urea nitrogen; C3, complement 3; C4, complement 4; Ca, cal- 82 cium; CH50, 50% hemolytic complement activity; Cl, chloride; Cr, creatinine; CRP, c-reac- 83 tive protein; ds-DNA, double-stranded deoxyribonucleic acid antibody; eGFR, estimated 84 glomerular filtration rate; Glu, glucose; γGTP, γ-glutamyltranspeptidase; Hb, hemoglo- 85 bin; HbA1c, hemoglobin a1c; HBV, hepatitis B virus; HCV, hepatitis C virus; HIV, human 86 immunodeficiency virus; IgA; immunoglobulin A, IgG, immunoglobulin G; IgM; immu- 87 noglobulin M; IP inorganic phosphate; K, kalium; LDH, lactate dehydrogenase; MPO- 88 ANCA, myeloperoxidase antineutrophil cytoplasmic antibody; Na, natrium; NAG, N-ac- 89 etyl-β-D-glucosaminidase; Plt, platelets; PR3-ANCA, proteinase3-antineutrophil cyto- 90 plasmic antibody; RBC, red blood cells; RF, rheumatoid factor; TP, total protein; UA, uric 91 acid; WBC, white blood cells. Mark this part as legend, I would omit a lot of »unimportant tests« because in a way what is now it is less transparent 

Response1:
Thank you for your insightful comment regarding the abbreviation list. We agree that a more concise list would improve the clarity of the manuscript. In response to your suggestion, we have removed the following abbreviations: β2MG (β2-microglobulin), Cryoglobulin, ASO (anti-streptolysin O antibody), RF (rheumatoid factor), Glu (glucose), UA (uric acid), Ca (calcium), and IP (inorganic phosphate). Additionally, we have moved the abbreviation list to a legend to further distinguish it from the main text. We believe this revised format now presents a more focused and relevant selection of abbreviations, enhancing the overall readability of the article.

Comment2:
Line 196,
The mentioned methods you can not compare. US and D-dimer are not the methods to diagnoze PE – so, please, use other formulation 

Response2:
We appreciate the reviewer's insightful comments. As suggested, we have revised the sentence to address the concern about comparing US and D-dimer, which are not diagnostic methods for PE.

Comment3:
Line 199 and on 
Discussion about V/Q scintigraphy is acually not used, you describe + and- of other methods and not of the method you used for diagnosis of PE; The main problem is , as you describe a lot of equivocal results. When you have already a changes on chst X ray the V/Q scinti is even less reliable. You have to talk about that and maybe not abopur other non used methods. 

Response3:
Thank you for your response. We have removed the discussion about V/Q scintigraphy as suggested and added a discussion of its limitations in the presence of pulmonary opacities.

Comment4:
Conclusions 
Line 239 While contrast-enhanced CT is the gold standard for diagnosis, V/Q scintigraphy re- 239 mains a valid diagnostic option, particularly in cases where renal dysfunction or other 240 factors preclude the use of contrast agents. Actually there is no contraindication for CTA in your patient, because renal function was not seriously diminished So it is difficult to defend your position. 

Response4:
Thank you for your comment. We agree that our patient did not have a contraindication for CTA. Therefore, we have revised the conclusion to reflect this.